# K-Net: Towards Unified Image Segmentation

**Wenwei Zhang**[1]  **Jiangmiao Pang**[2,4]  **Kai Chen**[3,4]  **Chen Change Loy**[1✉]

[1]S-Lab, Nanyang Technological University
[2]CUHK-SenseTime Joint Lab, the Chinese University of Hong Kong
[3]SenseTime Research   [4]Shanghai AI Laboratory
{wenwei001, ccloy}@ntu.edu.sg   pangjiangmiao@gmail.com
chenkai@sensetime.com

## Abstract

Semantic, instance, and panoptic segmentations have been addressed using different and specialized frameworks despite their underlying connections. This paper presents a unified, simple, and effective framework for these essentially similar tasks. The framework, named **K-Net**, segments both instances and semantic categories consistently by a group of learnable *kernels*, where each kernel is responsible for generating a mask for either a potential instance or a stuff class. To remedy the difficulties of distinguishing various instances, we propose a kernel update strategy that enables each kernel dynamic and conditional on its meaningful group in the input image. K-Net can be trained in an end-to-end manner with bipartite matching, and its training and inference are naturally NMS-free and box-free. Without bells and whistles, K-Net surpasses all previous published state-of-the-art single-model results of panoptic segmentation on MS COCO `test-dev` split and semantic segmentation on ADE20K val split with **55.2%** PQ and **54.3%** mIoU, respectively. Its instance segmentation performance is also on par with Cascade Mask R-CNN on MS COCO with 60%-90% faster inference speeds. Code and models will be released at `https://github.com/ZwwWayne/K-Net/`.

## 1 Introduction

Image segmentation aims at finding groups of coherent pixels [48]. There are different notions in groups, such as semantic categories (*e.g.*, car, dog, cat) or instances (*e.g.*, objects that coexist in the same image). Based on the different segmentation targets, the tasks are termed differently, *i.e.*, semantic and instance segmentation, respectively. There are also pioneer attempts [19, 29, 50, 62] to joint the two segmentation tasks for more comprehensive scene understanding.

Grouping pixels according to semantic categories can be formulated as a dense classification problem. As shown in Fig. 1-(a), recent methods directly learn a set of convolutional kernels (namely *semantic kernels* in this paper) of pre-defined categories and use them to classify pixels [40] or regions [22]. Such a framework is elegant and straightforward. However, extending this notion to instance segmentation is non-trivial given the varying number of instances across images. Consequently, instance segmentation is tackled by more complicated frameworks with additional steps such as object detection [22] or embedding generation [44]. These methods rely on extra components, which must guarantee the accuracy of extra components to a reasonable extent, or demand complex post-processing such as Non-Maximum Suppression (NMS) and pixel grouping. Recent approaches [34, 49, 55] generate kernels from dense feature grids and then select kernels for segmentation to simplify the frameworks. Nonetheless, since they build upon dense grids to enumerate and select kernels, these methods still rely on hand-crafted post-processing to eliminate masks or kernels of duplicated instances.

35th Conference on Neural Information Processing Systems (NeurIPS 2021).

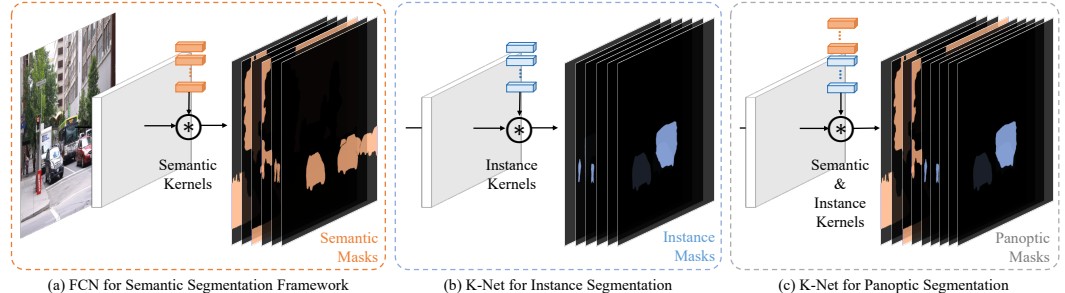

| (a) FCN for Semantic Segmentation Framework | (b) K-Net for Instance Segmentation | (c) K-Net for Panoptic Segmentation |

Figure 1: Semantic segmentation (a), instance (b), and panoptic segmentation (c) tasks are unified by a common framework in this paper. In conventional semantic segmentation methods, each convolutional kernel corresponds to a semantic class. Our framework extends this notion to make each kernel corresponds to either a potential instance or a semantic class.

In this paper, we make the first attempt to formulate a unified and effective framework to bridge the seemingly different image segmentation tasks (semantic, instance, and panoptic) through the notion of *kernels*. Our method is dubbed as K-Net ('K' stands for kernels). It begins with a set of convolutional kernels that are randomly initialized, and learns the kernels in accordance to the segmentation targets at hand, namely, *semantic kernels* for semantic categories and *instance kernels* for instance identities (Fig. 1-(b)). A simple combination of semantic kernels and instance kernels allows panoptic segmentation naturally (Fig. 1-(c)). In the forward pass, the kernels perform convolution on the image features to obtain the corresponding segmentation predictions.

The versatility and simplicity of K-Net are made possible through two designs. First, we formulate K-Net so that it dynamically updates the kernels to make them conditional to their activations on the image. Such a content-aware mechanism is crucial to ensure that each kernel, especially an instance kernel, responds accurately to varying objects in an image. Through applying this adaptive kernel update strategy iteratively, K-Net significantly improves the discriminative ability of the kernels and boosts the final segmentation performance. It is noteworthy that this strategy universally applies to kernels for all the segmentation tasks.

Second, inspired by recent advances in object detection [4], we adopt the bipartite matching strategy [47] to assign learning targets for each kernel. This training approach is advantageous to conventional training strategies [37, 46] as it builds a one-to-one mapping between kernels and instances in an image. It thus resolves the problem of dealing with a varying number of instances in an image. In addition, it is purely mask-driven without involving boxes. Hence, K-Net is naturally NMS-free and box-free, which is appealing to real-time applications.

To show the effectiveness of the proposed unified framework on different segmentation tasks, we conduct extensive experiments on COCO dataset [38] for panoptic and instance segmentation, and ADE20K dataset [70] for semantic segmentation. Without bells and whistles, K-Net surpasses all previous state-of-the-art single-model results on panoptic (**54.6%** PQ) and semantic segmentation benchmarks (**54.3%** mIoU) and achieves competitive performance compared to the more expensive Cascade Mask R-CNN [3]. We further analyze the learned kernels and find that instance kernels incline to specialize on objects at specific locations of similar sizes.

## 2 Related Work

**Semantic Segmentation.** Contemporary semantic segmentation approaches typically build upon a fully convolutional network (FCN) [40] and treat the task as a dense classification problem. Based on this framework, many studies focus on enhancing the feature representation through dilated convolution [7–9], pyramid pooling [57, 67], context representations [64, 65], and attention mechanisms [32, 63, 68]. Recently, SETR [69] reformulates the task as a sequence-to-sequence prediction task by using a vision transformer [17]. Despite the different model architectures, the approaches above share the common notion of making predictions via static semantic kernels. Differently, the proposed K-Net makes the kernels dynamic and conditional on their activations in the image.

**Instance Segmentation.** There are two representative frameworks for instance segmentation – 'top-down' and 'bottom-up' approaches. 'Top-down' approaches [2, 15, 22, 27, 33] first detect accurate

bounding boxes and generate a mask for each box. Mask R-CNN [22] simplifies this pipeline by directly adding a FCN [40] in Faster R-CNN [46]. Extensions of this framework add a mask scoring branch [24] or adopt a cascade structure [3, 5]. 'Bottom-up' methods [1, 30, 43, 44] first perform semantic segmentation then group pixels into different instances. These methods usually require a grouping process, and their performance often appears inferior to 'top-down' approaches in popular benchmarks [14, 38]. Unlike all these works, K-Net performs segmentation and instance separation simultaneously by constraining each kernel to predict one mask at a time for one object. Therefore, K-Net needs neither bounding box detection nor grouping process. It focuses on refining kernels rather than refining bounding boxes, different from previous cascade methods [3, 5].

Recent attempts [10, 54, 55, 58] perform instance segmentation in one stage without involving detection nor embedding generation. These methods apply dense mask prediction using dense sliding windows [10] or dense grids [54]. Some studies explore polar [58] representation, contour [45], and explicit shape representation [60] of instance masks. These methods all rely on NMS to eliminate duplicated instance masks, which hinders end-to-end training. The heuristic process is also unfavorable for real-time applications. Instance kernels in K-Net are trained in an end-to-end manner with bipartite matching and set prediction loss, thus, our methods does not need NMS.

**Panoptic Segmentation.** Panoptic segmentation [29] combines instance and semantic segmentation to provide a richer understanding of the scene. Different strategies have been proposed to cope with the instance segmentation task. Mainstream frameworks add a semantic segmentation branch [28, 31, 55, 59] on an instance segmentation framework or adopt different pixel grouping strategies [11, 61] based on a semantic segmentation method. Recently, DETR [4] tries to simplify the framework by transformer [51] but need to predict boxes around both stuff and things classes in training for assigning learning targets. These methods either need object detection or embedding generation to separate instances, which does not reconcile the instance and semantic segmentation in a unified framework. By contrast, K-Net partitions an image into semantic regions by semantic kernels and object instances by instance kernels through a unified perspective of kernels.

Concurrent to K-Net, some recent attempts [12, 35, 52] apply Transformer [51] for panoptic segmentation. MaskFormer reformulates semantic segmentation as a mask classification task, which is commonly adopted in instance-level segmentation. From an inverse perspective, K-Net tries to simplify instance and panoptic segmentation by letting a kernel to predict the mask of only one instance or a semantic category, which is the essential design in semantic segmentation. In contrast to K-Net that directly uses learned kernels to predict masks and progressively refines the masks and kernels, MaX-DeepLab [52] and MaskFormer [12] rely on queries and Transformer [51] to produce dynamic kernels for the final mask prediction.

**Dynamic Kernels.** Convolution kernels are usually static, *i.e.*, agnostic to the inputs, and thus have limited representation ability. Previous works [16, 18, 25, 26, 71] explore different kinds of dynamic kernels to improve the flexibility and performance of models. Some semantic segmentation methods apply dynamic kernels to improve the model representation with enlarged receptive fields [56] or multi-scales contexts [20]. Differently, K-Net uses dynamic kernels to improve the discriminative capability of the segmentation kernels more so than the input features of kernels.

Recent studies apply dynamic kernels to generate instance [49, 55] or panoptic [34] segmentation predictions directly. Because these methods generate kernels from dense feature maps, enumerate kernels of each position, and filter out kernels of background regions, they either still rely on NMS [49, 55] or need extra kernel fusion [34] to eliminate kernels or masks of duplicated objects. Instead of generated from dense grids, the kernels in K-Net are a set of learnable parameters updated by their corresponding contents in the image. K-Net does not need to handle duplicated kernels because its kernels learn to focus on different regions of the image in training, constrained by the bipartite matching strategy that builds a one-to-one mapping between the kernels and instances.

## 3   Methodology

We consider various segmentation tasks through a unified perspective of kernels. The proposed K-Net uses a set of kernels to assign each pixel to either a potential instance or a semantic class (Sec. 3.1). To enhance the discriminative capability of kernels, we contribute a way to update the static kernels by the contents in their partitioned pixel groups (Sec. 3.2). We adopt the bipartite

matching strategy to train instance kernels in an end-to-end manner (Sec. 3.3). K-Net can be applied seamlessly to semantic, instance, and panoptic segmentation as described in Sec. 3.4.

## 3.1 K-Net

Despite the different definitions of a 'meaningful group', all segmentation tasks essentially assign each pixel to one of the predefined meaningful groups [48]. As the number of groups in an image is typically assumed finite, we can set the maximum group number of a segmentation task as $N$. For example, there are $N$ pre-defined semantic classes for semantic segmentation or at most $N$ objects in an image for instance segmentation. For panoptic segmentation, $N$ is the total number of stuff classes and objects in an image. Therefore, we can use $N$ kernels to partition an image into $N$ groups, where each kernel is responsible to find the pixels belonging to its corresponding group. Specifically, given an input feature map $F \in R^{B \times C \times H \times W}$ of $B$ images, produced by a deep neural network, we only need $N$ kernels $K \in R^{N \times C}$ to perform convolution with $F$ to obtain the corresponding segmentation prediction $M \in R^{B \times N \times H \times W}$ as

$$M = \sigma(K * F), \tag{1}$$

where $C$, $H$, and $W$ are the number of channels, height, and width of the feature map, respectively. The activation function $\sigma$ can be softmax function if we want to assign each pixel to only one of the kernels (usually used in semantic segmentation). Sigmoid function can also be used as activation function if we allow one pixel belong to multiple masks, which results on $N$ binary masks by setting a threshold like 0.5 on the activation map (usually used in instance segmentation).

This formulation has already dominated semantic segmentation for years [8, 40, 67]. In semantic segmentation, each kernel is responsible to find all pixels of a similar class across images. Whereas in instance segmentation, each pixel group corresponds to an object. However, previous methods separate instances by extra steps [22, 30, 44] instead of by kernels.

This paper is the first study that explores if the notion of kernels in semantic segmentation is equally applicable to instance segmentation, and more generally panoptic segmentation. To separate instances by kernels, each kernel in K-Net only segments at most one object in an image (Fig. 1-(b)). In this way, K-Net distinguishes instances and performs segmentation simultaneously, achieving instance segmentation in one pass without extra steps. For simplicity, we call these kernels as *semantic* and *instance* kernels in this paper for semantic and instance segmentation, respectively. A simple combination of instance kernels and semantic kernels can naturally preform panoptic segmentation that either assigns a pixel to an instance ID or a class of stuff (Fig. 1-(c)).

## 3.2 Group-Aware Kernels

Despite the simplicity of K-Net, separating instances directly by kernels is non-trivial. Because instance kernels need to discriminate objects that vary in scale and appearance within and across images. Without a common and explicit characteristic like semantic categories, the instance kernels need stronger discriminative ability than static kernels.

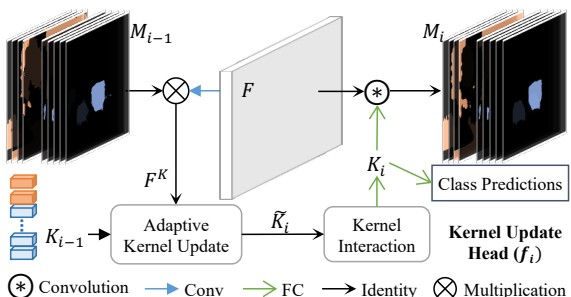

Figure 2: Kernel Update Head.

To overcome this challenge, we contribute an approach to make the kernel conditional on their corresponding pixel groups, through a kernel update head, as shown in Fig. 2. The kernel update head $f_i$ contains three key steps: group feature assembling, adaptive kernel update, and kernel interaction. Firstly, the group feature $F^K$ for each pixel group is assembled using the mask prediction $M_{i-1}$. As it is the content of each individual groups that distinguishes them from each other, $F^K$ is used to update their corresponding kernel $K_{i-1}$ adaptively. After that, the kernel interacts with each other to comprehensively model the image context. Finally, the obtained group-aware kernels $K_i$ perform convolution over feature map $F$ to obtain more accurate mask prediction $M_i$. As shown in Fig. 3, this process can be conducted iteratively because a finer partition usually reduces the noise in group features, which results in more discriminative kernels. This process is formulated as

$$K_i, M_i = f_i(M_{i-1}, K_{i-1}, F). \tag{2}$$

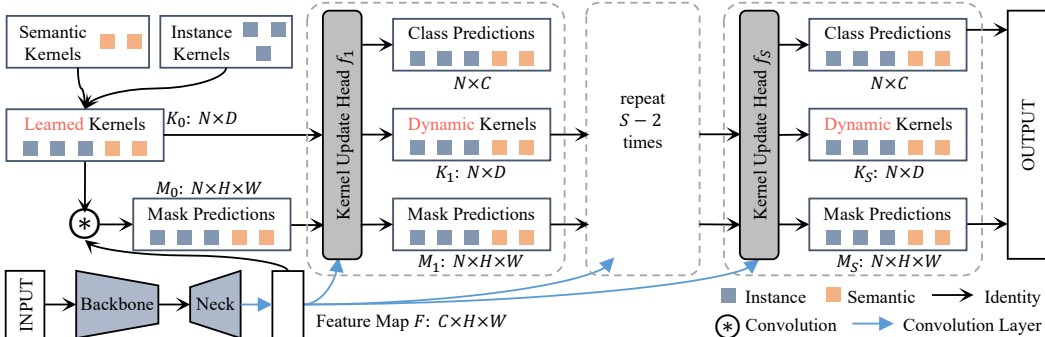

Figure 3: **K-Net for panoptic segmentaion.** A set of learned kernels first performs convolution with the feature map $F$ to predict masks $M_0$. Then the kernel update head takes the mask predictions $M_0$, learned kernels $K_0$, and feature map $F$ as input and produce class predictions, group-aware (dynamic) kernels, and mask predictions. The produced mask prediction, dynamic kernels, and feature map $F$ are sent to the next kernel update head. This process is performed iteratively to progressively refine the kernels and the mask predictions.

Notably, the kernel update head with the iterative refinement is universal as it does not rely on the characteristic of kernels. Thus, it can enhance not only instance kernels but also semantic kernels. We detail the three steps as follows.

**Group Feature Assembling.** The kernel update head first assembles the features of each group, which will be adopted later to make the kernels group-aware. As the mask of each kernel in $M_{i-1}$ essentially defines whether or not a pixel belongs to the kernel's related group, we can assemble the feature $F^K$ for $K_{i-1}$ by multiplying the feature map $F$ with the $M_{i-1}$ as

$$F^K = \sum_u^H \sum_v^W M_{i-1}(u,v) \cdot F(u,v), F^K \in R^{B \times N \times C}, \tag{3}$$

where $B$ is the batch size, $N$ is the number of kernels, and $C$ is the number of channels.

**Adaptive Feature Update.** The kernel update head then updates the kernels using the obtained $F^K$ to improve the representation ability of kernels. As the mask $M_{i-1}$ may not be accurate, which is more common the case, the feature of each group may also contain noises introduced by pixels from other groups. To reduce the adverse effect of the noise in group features, we devise an adaptive kernel update strategy. Specifically, we first conduct element-wise multiplication between $F^K$ and $K_{i-1}$ as

$$F^G = \phi_1(F^K) \otimes \phi_2(K_{i-1}), F^G \in R^{B \times N \times C}, \tag{4}$$

where $\phi_1$ and $\phi_2$ are linear transformations. Then the head learns two gates, $G^F$ and $G^K$, which adapt the contribution from $F^K$ and $K_{i-1}$ to the updated kernel $\tilde{K}$, respectively. The formulation is

$$G^K = \sigma(\psi_1(F^G)), G^F = \sigma(\psi_2(F^G)),$$
$$\tilde{K} = G^F \otimes \psi_3(F^K) + G^K \otimes \psi_4(K_{i-1}), \tag{5}$$

where $\psi_n, n = 1, ..., 4$ are different fully connected (FC) layers followed by LayerNorm (LN) and $\sigma$ is the Sigmoid function. $\tilde{K}$ is then used in kernel interaction.

The gate learned here plays a role like the self-attention mechanism in Transformer [51], whose output is computed as a weighted summation of the values. In Transformer, the weight assigned to each value is usually computed by a compatibility function dot-product of the queries and keys. Similarly, adaptive kernel update essentially performs weighted summation of kernel features $K_{i-1}$ and group features $F^G$. Their weight $G^K$ and $G^F$ are computed by element-wise multiplication, which can be regarded as another kind of compatibility function.

**Kernel Interaction.** Interaction among kernels is important to inform each kernel with contextual information from other groups. Such information allows the kernel to implicitly model and exploit the relationship between groups of an image. To this end, we add a kernel interaction process to obtain the new kernels $K_i$ given the updated kernels $\tilde{K}$. Here we simply adopt Multi-Head Attention [51] followed by a Feed-Forward Neural Network, which has been proven effective in previous works [4, 51]. The output $K_i$ of kernel interaction is then used to generate a new mask prediction through $M_i = g_i(K_i) * F$, where $g_i$ is an FC-LN-ReLU layer followed by an FC layer. $K_i$ will also be used to predict classification scores in instance and panoptic segmentation.

### 3.3 Training Instance Kernels

While each semantic kernel can be assigned to a constant semantic class, there lacks an explicit rule to assign varying number of targets to instance kernels. In this work, we adopt bipartite matching strategy and set prediction loss [4, 47] to train instance kernels in an end-to-end manner. Different from previous works [4, 47] that rely on boxes, the learning of instance kernels is purely mask-driven because the inference of K-Net is naturally box-free.

**Loss Functions.** The loss function for instance kernels is written as $L_K = \lambda_{cls}L_{cls} + \lambda_{ce}L_{ce} + \lambda_{dice}L_{dice}$, where $L_{cls}$ is Focal loss [37] for classification, and $L_{ce}$ and $L_{dice}$ are CrossEntropy (CE) loss and Dice loss [42] for segmentation, respectively. Given that each instance only occupies a small region in an image, CE loss is insufficient to handle the highly imbalanced learning targets of masks. Therefore, we apply Dice loss [42] to handle this issue following previous works [49, 54, 55].

**Mask-based Hungarian Assignment.** We adopt Hungarian assignment strategy used in [4, 47] for target assignment to train K-Net in an end-to-end manner. It builds a one-to-one mapping between the predicted instance masks and the ground-truth (GT) instances based on the matching costs. The matching cost is calculated between the mask and GT pairs in a similar manner as the training loss.

### 3.4 Applications to Various Segmentation Tasks

**Panoptic Segmentation.** For panoptic segmentation, the kernels are composed of instance kernels $K_0^{ins}$ and semantic kernels $K_0^{sem}$ as shown in Fig. 3. We adopt semantic FPN [28] for producing high resolution feature map $F$, except that we add positional encoding used in [4, 51, 72] to enhance the positional information. Specifically, given the feature maps $P2, P3, P4, P5$ produced by FPN [36], positional encoding is computed based on the feature map size of $P5$, and it is added with $P5$. Then semantic FPN [28] is used to produce the final feature map.

As semantic segmentation mainly relies on semantic information for per-pixel classification, while instance segmentation prefers accurate localization information to separate instances, we use two separate branches to generate the features $F^{ins}$ and $F^{sem}$ to perform convolution with $K_0^{ins}$ and $K_0^{sem}$ for generating instance and semantic masks $M_0^{ins}$ and $M_0^{sem}$, respectively. Notably, it is unnecessary to produce 'thing' and 'stuff' masks initially from different branches to produce a reasonable performance. Such a design is consistent with previous practices [28, 55] and empirically yields better performance (about 1% PQ).

We then construct $M_0$, $K_0$, and $F$ as the inputs of kernel update head to dynamically update the kernels and refine the panoptic mask prediction. Because 'things' are already separated by instance masks in $M_0^{ins}$, while $M_0^{sem}$ contains the semantic masks of both 'things' and 'stuff', we select $M_0^{st}$, the masks of stuff categories from $M_0^{sem}$, and directly concatenate it with $M_0^{ins}$ to form the panoptic mask prediction $M_0$. Due to similar reason, we only select and concatenate the kernels of stuff classes in $K_0^{sem}$ with $K_0^{ins}$ to form the panoptic kernels $K_0$. To exploit the complementary semantic information in $F^{sem}$ and localization information in $F^{ins}$, we add them together to obtain $F$ as the input feature map of the kernel update head. With $M_0$, $K_0$, and $F$, the kernel update head $f_1$ can produce group-aware kernels $K_1$ and mask $M_1$. Then kernels and masks are iteratively by $S$ times and finally we can obtain the mask prediction $M_S$.

To produce the final panoptic segmentation results, we paste thing and stuff masks in a mixed order following MaskFormer [12]. We also find it necessary in K-Net to firstly sort the pasting order of masks based on their classification scores for further filtering out lower-confident mask predictions. Such a method empirically performs better (about 1% PQ) than the previous strategy that pasting thing and stuff masks separately [28, 34].

**Instance Segmentation.** In the similar framework, we simply remove the concatenation process of kernels and masks to perform instance segmentation. We did not remove the semantic segmentation branch as the semantic information is still complementary for instance segmentation. Note that in this case, the semantic segmentation branch does not use extra annotations. The ground truth of semantic segmentation is built by converting instance masks to their corresponding class labels.

**Semantic Segmentation.** As K-Net does not rely on specific architectures of model representation, K-Net can perform semantic segmentation by simply appending its kernel update head to any existing semantic segmentation methods [8, 40, 57, 67] that rely on semantic kernels.

Table 1: Comparisons with state-of-the-art panoptic segmentation methods on COCO dataset

| Framework | Backbone | Box-free | NMS-free | Epochs | PQ | PQ$^{Th}$ | PQ$^{St}$ |
|---|---|---|---|---|---|---|---|
| val | | | | | | | |
| Panoptic-DeepLab [11] | Xception-71 | | | ~1000 | 39.7 | 43.9 | 33.2 |
| Panoptic FPN [28] | R50-FPN | | | 36 | 41.5 | 48.5 | 31.1 |
| SOLOv2 [55] | R50-FPN | ✓ | | 36 | 42.1 | 49.6 | 30.7 |
| DETR [4]† | R50 | | ✓ | 300 + 25 | 43.4 | 48.2 | 36.3 |
| Unifying [31] | R50-FPN | | | ~27 | 43.4 | 48.6 | 35.5 |
| Panoptic FCN [34] | R50-FPN | ✓ | ✓ | 36 | 43.6 | 49.3 | 35.0 |
| K-Net | R50-FPN | ✓ | ✓ | 36 | **47.1** | **51.7** | **40.3** |
| | R101-FPN | ✓ | ✓ | 36 | 49.6 | 55.1 | 41.4 |
| K-Net | R101-FPN-DCN | ✓ | ✓ | 36 | 48.3 | 54.0 | 39.7 |
| | Swin-L [39] | ✓ | ✓ | 36 | 54.6 | 60.2 | 46.0 |
| test-dev | | | | | | | |
| Panoptic-DeepLab | Xception-71 | ✓ | | ~1000 | 41.4 | 45.1 | 35.9 |
| Panoptic FPN | R101-FPN | | | 36 | 43.5 | 50.8 | 32.5 |
| Panoptic FCN | R101-FPN | ✓ | ✓ | 36 | 45.5 | 51.4 | 36.4 |
| DETR | R101 | | ✓ | 300 + 25 | 46.0 | - | - |
| UPSNet [59] | R101-FPN-DCN | | | 36 | 46.6 | 53.2 | 36.7 |
| Unifying [31] | R101-FPN-DCN | | | ~27 | 47.2 | 53.5 | 37.7 |
| K-Net | R101-FPN | ✓ | ✓ | 36 | 47.0 | 52.8 | 38.2 |
| K-Net | R101-FPN-DCN | ✓ | ✓ | 36 | **48.3** | **54.0** | **39.7** |
| MaX-DeepLab-L [52] | Max-L | ✓ | ✓ | 54 | 51.3 | 57.2 | 42.4 |
| MaskFormer [12] | Swin-L [39] | ✓ | ✓ | 300 | 53.3 | 59.1 | 44.5 |
| Panoptic SegfFormer [35] | PVTv2-B5 [53] | ✓ | ✓ | 50 | 54.4 | 61.1 | 44.3 |
| K-Net | Swin-L | ✓ | ✓ | 36 | **55.2** | **61.2** | **46.2** |

## 4  Experiments

**Dataset and Metrics.** For panoptic and instance segmentation, we perform experiments on the challenging COCO dataset [38]. All models are trained on the `train2017` split and evaluated on the `val2017` split. The panoptic segmentation results are evaluated by the PQ metric [29]. We also report the performance of thing and stuff, noted as PQ$^{Th}$, PQ$^{St}$, respectively, for thorough evaluation. The instance segmentation results are evaluated by mask AP [38]. The AP for small, medium and large objects are noted as AP$_s$, AP$_m$, and AP$_l$, respectively. The AP at mask IoU thresholds 0.5 and 0.75 are also reported as AP$_{50}$ and AP$_{75}$, respectively. For semantic segmentation, we conduct experiments on the challenging ADE20K dataset [70] and report mIoU to evaluate the segmentation quality. All models are trained on the `train` split and evaluated on the `validation` split.

**Implementation Details.** For panoptic and instance segmentation, we implement K-Net with MMDetection [6]. In the ablation study, the model is trained with a batch size of 16 for 12 epochs. The learning rate is 0.0001, and it is decreased by 0.1 after 8 and 11 epochs, respectively. We use AdamW [41] with a weight decay of 0.05. For data augmentation in training, we adopt horizontal flip augmentation with a single scale. The long edge and short edge of images are resized to 1333 and 800, respectively, without changing the aspect ratio. When comparing with other frameworks, we use multi-scale training with a longer schedule (36 epochs) for fair comparisons [6]. The short edge of images is randomly sampled from $[640, 800]$ [21].

For semantic segmentation, we implement K-Net with MMSegmentation [13] and train it with 80,000 iterations. As AdamW [41] empirically works better than SGD, we use AdamW with a weight decay of 0.0005 by default on both the baselines and K-Net for a fair comparison. The initial learning rate is 0.0001, and it is decayed by 0.1 after 60000 and 72000 iterations, respectively. More details are provided in the appendix.

**Model Hyperparameters.** In the ablation study, we adopt ResNet-50 [23] backbone with FPN [36]. For panoptic and instance segmentation, we use $\lambda_{cls} = 2$ for Focal loss following previous methods [72], and empirically find $\lambda_{seg} = 1$, $\lambda_{ce} = 1$, $\lambda_{dice} = 4$ work best. For efficiency, the default number of instance kernels is 100. For semantic segmentation, $N$ equals to the number of classes of the dataset, which is 150 in ADE20K and 133 in COCO dataset. The number of rounds of iterative kernel update is set to three by default for all segmentation tasks.

### 4.1  Benchmark Results

**Panoptic Segmentation.** We first benchmark K-Net with other panoptic segmentation frameworks in Table 1. K-Net surpasses the previous state-of-the-art box-based method [31] and box/NMS-free method [34] by 1.7 and 1.5 PQ on `val` split, respectively. On the `test-dev` split, K-Net with ResNet-

Table 2: Comparisons with state-of-the-art instance segmentation methods on COCO dataset. 'P. (M)' indicates the number of parameters in the model, and the counting unit is million

| Method | Backbone | Box-free | NMS-free | Epochs | AP↑ | $AP_{50}$ | $AP_{70}$ | $AP_s$ | $AP_m$ | $AP_l$ | FPS↑ | P. (M)↓ |
|---|---|---|---|---|---|---|---|---|---|---|---|---|
| *val2017* | | | | | | | | | | | | |
| SOLO [54] | R-50-FPN | ✓ | | 36 | 35.8 | 56.7 | 37.9 | 14.3 | 39.3 | 53.2 | 12.7 | 36.08 |
| Mask R-CNN [22] | R-50-FPN | | | 36 | 37.1 | 58.5 | 39.7 | 18.7 | 39.6 | 53.9 | 17.5 | 44.17 |
| SOLOv2 [55] | R-50-FPN | ✓ | | 36 | 37.5 | 58.2 | 40.0 | 15.8 | 41.4 | 56.6 | 17.7 | **33.89** |
| CondInst [49] | R-50-FPN | | | 36 | 37.5 | 58.5 | 40.1 | 18.7 | 41.0 | 53.3 | 14.0 | 46.37 |
| Cascade Mask R-CNN [3] | R-50-FPN | | | 36 | 38.5 | 59.7 | **41.8** | **19.3** | 41.1 | 55.6 | 10.3 | 77.10 |
| K-Net | R-50-FPN | ✓ | ✓ | 36 | 37.8 | 60.3 | 39.9 | 16.9 | 41.2 | 57.5 | **21.2** | 37.26 |
| K-Net-N256 | R-50-FPN | ✓ | ✓ | 36 | **38.6** | **60.9** | 41.0 | 19.1 | **42.0** | **57.7** | 19.8 | 37.30 |
| *test-dev* | | | | | | | | | | | | |
| SOLO | R-50-FPN | ✓ | | 72 | 36.8 | 58.6 | 39.0 | 15.9 | 39.5 | 52.1 | 12.7 | 36.08 |
| Mask R-CNN | R-50-FPN | | | 36 | 37.4 | 59.5 | 40.1 | 18.6 | 39.8 | 51.6 | 17.5 | 44.17 |
| CondInst | R-50-FPN | | | 36 | 37.8 | 59.2 | 40.4 | 18.2 | 40.3 | 52.7 | 14.0 | 46.37 |
| SOLOv2 | R-50-FPN | ✓ | | 36 | 38.2 | 59.3 | 40.9 | 16.0 | 41.2 | 55.4 | 17.7 | **33.89** |
| Cascade Mask R-CNN | R-50-FPN | | | 36 | 38.8 | 60.4 | **42.0** | **19.4** | 40.9 | 53.9 | 10.3 | 77.10 |
| K-Net | R-50-FPN | ✓ | ✓ | 36 | 38.4 | 61.2 | 40.9 | 17.4 | 40.7 | 56.2 | **21.2** | 37.26 |
| K-Net-N256 | R-50-FPN | ✓ | ✓ | 36 | **39.1** | **61.7** | 41.8 | 18.2 | **41.4** | **56.6** | 19.8 | 37.30 |
| SOLO | R-101-FPN | ✓ | | 72 | 37.8 | 59.5 | 40.4 | 16.4 | 40.6 | 54.2 | 10.7 | 55.07 |
| Mask R-CNN | R-101-FPN | | | 36 | 38.8 | 60.8 | 41.8 | 19.1 | 41.2 | 54.3 | 14.3 | 63.16 |
| CondInst | R-101-FPN | | | 36 | 38.9 | 60.6 | 41.8 | 18.8 | 41.8 | 54.4 | 11.0 | **52.83** |
| SOLOv2 | R-101-FPN | ✓ | | 36 | 39.5 | 60.8 | 42.6 | 16.7 | 43.0 | 57.4 | 14.3 | 65.36 |
| Cascade Mask R-CNN | R-101-FPN | | | 36 | 39.9 | 61.6 | 43.3 | **19.8** | 42.1 | 55.7 | 9.5 | 96.09 |
| K-Net | R-101-FPN | ✓ | ✓ | 36 | 40.1 | 62.8 | 43.1 | 18.7 | 42.7 | 58.8 | **16.2** | 56.25 |
| K-Net-N256 | R-101-FPN | ✓ | ✓ | 36 | **40.6** | **63.3** | **43.7** | 18.8 | **43.3** | **59.0** | 15.5 | 56.29 |

Table 3: Results of K-Net on ADE20K semantic segmentation dataset

(a) Improvements of K-Net on different architectures

| Method | Backbone | Val mIoU |
|---|---|---|
| FCN [40] | R50 | 36.7 |
| FCN + K-Net | R50 | 43.3 *(+6.6)* |
| PSPNet [67] | R50 | 42.6 |
| PSPNet + K-Net | R50 | 43.9 *(+1.3)* |
| DLab.v3 [8] | R50 | 43.5 |
| DLab.v3 + K-Net | R50 | 44.6 *(+1.1)* |
| UperNet [57] | R50 | 42.4 |
| UperNet + K-Net | R50 | 43.6 *(+1.2)* |
| UperNet | Swin-L | 50.6 |
| UperNet + K-Net | Swin-L | 52 *(+1.4)* |

(b) Comparisons with state-of-the-art methods. Results marked by † use larger image sizes

| Method | Backbone | Val mIoU |
|---|---|---|
| OCRNet [64] | HRNet-W48 | 44.9 |
| PSPNet [67] | R101 | 45.4 |
| PSANet [68] | R101 | 45.4 |
| DNL [63] | R101 | 45.8 |
| DLab.v3 [8] | R101 | 46.7 |
| DLab.v3+ [9] | S-101 [66] | 47.3 |
| SETR [69] | ViT-L [17] | 48.6 |
| UperNet† | Swin-L | 53.5 |
| UperNet + K-Net | Swin-L | 53.3 |
| UperNet + K-Net† | Swin-L | **54.3** |

101-FPN backbone even obtains better results than that of UPSNet [59], which uses Deformable Convolution Network (DCN) [16]. K-Net equipped with DCN surpasses the previous method [31] by 1.1 PQ. Without bells and whistles, K-Net obtains new state-of-the-art single-model performance with Swin Transformer [39] serving as the backbone.

We also compare K-Net with concurrent work Max-DeepLab [52], MaskFormer [12], and Panoptic SegFormer [35]. K-Net surpasses these methods with the least training epochs (36), taking only about 44 GPU days (roughly 2 days and 18 hours with 16 GPUs). Note that only 100 instance kernels and Swin Transformer with window size 7 are used here for efficiency. K-Net could obtain a higher performance with more instance kernels (Sec. 4.2), Swin Transformer with window size 12 (used in MaskFormer [12]), as well as an extended training schedule with aggressive data augmentation used in previous work [4].

**Instance Segmentation.** We compare K-Net with other instance segmentation frameworks [10, 22, 49] in Table 2. More details are provided in the appendix. As the only box-free and NMS-free method, K-Net achieves better performance and faster inference speed than Mask R-CNN [22], SOLO [54], SOLOv2 [55] and CondInst [49], indicated by the higher AP and frames per second (FPS). We adopt 256 instance kernels (K-Net-N256 in the table) to compare with Cascade Mask R-CNN [3]. The performance of K-Net-N256 is on par with Cascade Mask R-CNN [3] but enjoys a **92.2%** faster inference speed (19.8 v.s 10.3).

On COCO `test-dev` split, K-Net with ResNet-101-FPN backbone obtains performance that is 0.9 AP better than Mask R-CNN [22]. It also surpasses previous kernel-based approach CondInst [49] and SOLOv2 [55] by 1.2 AP and 0.6 AP, respectively. With ResNet-101-FPN backbone, K-Net surpasses Cascade Mask R-CNN with 100 and 256 instance kernels in both accuracy and speed by 0.2 AP and 6.7 FPS, and 0.7 AP and 6 FPS, respectively.

Table 4: Ablation studies of K-Net on instance segmentation

(a) Adaptive Kernel Update (A. K. U.) and Kernel Interaction (K. I.)

| A. K. U. | K. I. | AP | $AP_{50}$ | $AP_{75}$ |
|---|---|---|---|---|
| | | 10.0 | 18.2 | 9.6 |
| ✓ | | 22.6 | 37.3 | 23.5 |
| | ✓ | 31.2 | 52.0 | 32.4 |
| ✓ | ✓ | 34.1 | 55.3 | 35.7 |

(b) Positional Encoding (P. E.) and Coordinate Convolution (Coors.)

| Coors. | P. E. | AP | $AP_{50}$ | $AP_{75}$ |
|---|---|---|---|---|
| | | 30.9 | 51.7 | 31.6 |
| ✓ | | 34.0 | 55.4 | 35.6 |
| | ✓ | 34.1 | 55.3 | 35.7 |
| ✓ | ✓ | 34.0 | 55.1 | 35.8 |

(c) Number of rounds of kernel update

| Stage Number | AP | $AP_{50}$ | $AP_{75}$ | FPS |
|---|---|---|---|---|
| 1 | 21.8 | 37.3 | 22.1 | 24.0 |
| 2 | 32.1 | 52.3 | 33.5 | 22.7 |
| 3 | 34.1 | 55.3 | 35.7 | 21.2 |
| 4 | 34.5 | 56.5 | 35.7 | 20.1 |
| 5 | 34.5 | 56.5 | 35.9 | 18.9 |

(d) Numbers of instance kernels

| $N$ | AP | $AP_{50}$ | $AP_{75}$ | FPS |
|---|---|---|---|---|
| 50 | 32.7 | 53.7 | 34.1 | 21.6 |
| 64 | 33.6 | 54.8 | 35.1 | 21.6 |
| 100 | 34.1 | 55.3 | 35.7 | 21.2 |
| 128 | 34.3 | 55.6 | 35.8 | 20.7 |
| 256 | 34.7 | 56.1 | 36.3 | 19.8 |

We also compare the number of parameters of these models in Table 2. Though K-Net does not have the least number of parameters, it is more lightweight than Cascade Mask R-CNN by approximately half number of the parameters (37.3 M vs. 77.1 M).

**Semantic Segmentation.** We apply K-Net to existing frameworks [8, 40, 57, 67] that rely on static semantic kernels in Table 3a. K-Net consistently improves different frameworks. Notably, K-Net significantly improves FCN (**6.6** mIoU). This combination surpasses PSPNet and UperNet by 0.7 and 0.9 mIoU, respectively, and achieves performance comparable with DeepLab v3. Furthermore, the effectiveness of K-Net does not saturate with strong model representation, as it still brings significant improvement (1.4 mIoU) over UperNet with Swin Transformer [39]. The results suggest the versatility and effectiveness of K-Net for semantic segmentation.

In Table 3b, we further compare K-Net with other state-of-the-art methods [9, 69] with test-time augmentation on the validation set. With the input of 512×512, K-Net already achieves state-of-the-art performance. With a larger input of 640×640 following previous method [39] during training and testing, K-Net with UperNet and Swin Transformer achieves new state-of-the-art single model performance, which is 0.8 mIoU higher than the previous one.

## 4.2 Ablation Study on Instance Segmentation

We conduct an ablation study on COCO instance segmentation dataset to evaluate the effectiveness of K-Net in discriminating instances. The conclusion is also applicable to other segmentation tasks since the design of K-Net is universal to all segmentation tasks.

**Head Architecture.** We verify the components in the kernel update head in Table 4a. The results of without A. K. U. is obtained by updating kernels purely by $\tilde{K} = F_K + K_{i-1}$ followed by an FC-LN-ReLU layer. The results indicates that both adaptive kernel update and kernel interaction are necessary for high performance.

**Positional Information.** We study the necessity of positional information in Table 4b. The results show that positional information is beneficial, and positional encoding [4, 51] works slightly better than coordinate convolution. The combination of the two components does not bring additional improvements. The results justify the use of just positional encoding in our framework.

**Number of Stages.** We compare different kernel update rounds in Table 4c. The results show that FPS decreases as the update rounds grow while the performance saturates beyond three stages. Such a conclusion also holds for semantic segmentation as shown in Table 5. The performance of FCN + K-Net on ADE20K dataset gradually increases as the increase of iteration number but also saturates after four iterations.

Table 5: Numbers of semantic kernels

| Stage Number | 0 | 1 | 2 | 3 | 4 | 5 | 6 | 7 |
|---|---|---|---|---|---|---|---|---|
| mIoU | 36.7 | 42.7 | 43.0 | 43.3 | 43.8 | 44.1 | 43.1 | 42.6 |

**Number of Kernels.** We further study the number of kernels in K-Net. The results in Table 4d reveal that 100 kernels are sufficient to achieve good performance. The observation is expected for COCO dataset because most of the images in the dataset do not contain many objects (7.7 objects per image in average [38]). K-Net consistently achieves better performance given more instance kernels since they improve the models' capacity in coping with complicated images. However, a larger $N$ may lead

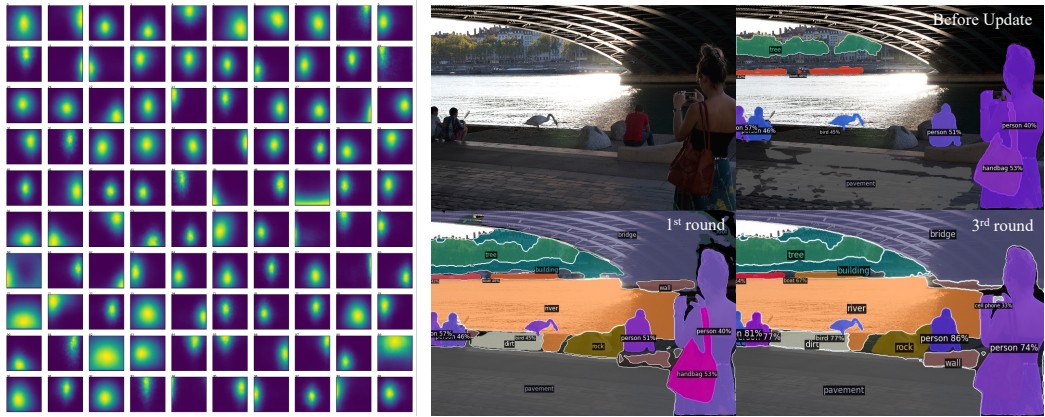

(a) Average activation over 5000 images.     (b) Mask prediction before and after kernel update.

Figure 4: Visual analysis of kernels and their masks. Best viewed in color and by zooming in.

to small performance gains and then get saturated (when $N = 300, 512, 768$, we all get 34.9% mAP). Therefore, we select $N = 100$ in other experiments for efficiency if without further specification.

### 4.3 Visual Analysis

**Overall Distribution of Kernels.** We carefully analyze the properties of instance kernels learned in K-Net by analyzing the average of mask activations of the 100 instance kernels over the 5000 images in the `val` split. All the masks are resized to have a similar resolution of $200 \times 200$ for the analysis. As shown in Fig. 4a, the learned kernels are meaningful. Different kernels specialize on different regions of the image and objects with different sizes, while each kernel attends to objects of similar sizes at close locations across images.

**Masks Refined through Kernel Update.** We further analyze how the mask predictions of kernels are refined through the kernel update in Fig. 4b. Here we take K-Net for panoptic segmentation to thoroughly analyze both semantic and instance masks. The masks produced by static kernels are incomplete, *e.g.*, the masks of river and building are missed. After kernel update, the contents are thoroughly covered by the segmentation masks, though the boundaries of masks are still unsatisfactory. The boundaries are refined after more kernel update. The classification confidences of instances also increase after kernel update. More results are given in the appendix.

## 5 Conclusion

This paper explores instance kernels that can learn to separate instances during segmentation. Thus, extra components that previously assist instance segmentation can be replaced by instance kernels, including bounding boxes, embedding generation, and hand-crafted post-processing like NMS, kernel fusion, and pixel grouping. Such an attempt, for the first time, allows different image segmentation tasks to be tackled through a unified framework. The framework, dubbed as K-Net, first partitions an image into different groups by learned static kernels, then iteratively refines these kernels and their partition of the image by the features assembled from their partitioned groups. K-Net obtains new state-of-the-art single-model performance on panoptic and semantic segmentation benchmarks and surpasses the well-developed Cascade Mask R-CNN with the fastest inference speed among the recent instance segmentation frameworks. We wish K-Net and the analysis to pave the way for future research on unified image segmentation frameworks.

**Acknowledgements.** This study is supported under the RIE2020 Industry Alignment Fund Industry Collaboration Projects (IAF-ICP) Funding Initiative, as well as cash and in-kind contribution from the industry partner(s). It is also partially supported by the NTU NAP grant. Jiangmiao Pang and Kai Chen are also supported by the Shanghai Committee of Science and Technology, China (Grant No. 20DZ1100800). The authors would like to thank the valuable suggestions and comments by Jiaqi Wang, Rui Xu, and Xingxing Zou.

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
