# K-Net: Towards Unified Image Segmentation Supplementary Materials

**Wenwei Zhang**[1]    **Jiangmiao Pang**[2,4]    **Kai Chen**[3,4]    **Chen Change Loy**[1⊠]

[1]S-Lab, Nanyang Technological University
[2]CUHK-SenseTime Joint Lab, the Chinese University of Hong Kong
[3]SenseTime Research    [4]Shanghai AI Laboratory
{wenwei001, ccloy}@ntu.edu.sg   pangjiangmiao@gmail.com
chenkai@sensetime.com

We first provide more implementation details for the three segmentation tasks of K-Net (Appendix A). Then we provide more benchmark details and discussion of comparison between K-Net and other methods (Appendix B). We further analyze K-Net about its results of kernel update and failure cases (Appendix C). Last but not the least, we discuss the broader impact of K-Net (Appendix D).

## A    Implementation Details

**Training details for semantic segmentation.** We implement K-Net based on MMSegmentation [4] for experiments on semantic segmentation. We use AdamW [7] with a weight decay of 0.0005 and train the model by 80000 iterations by default. The initial learning rate is 0.0001, and it is decayed by 0.1 after 60000 and 72000 iterations, respectively. This is different from the default training setting in MMSegmentation [4] that uses SGD with momentum by 160000 iterations. But our setting obtains similar performance as the default one. Therefore, we apply AdamW with 80000 iterations to all the models in Table 3a of the main text for efficiency while keeping fair comparisons. For data augmentation, we follow the default settings in MMSegmentation [4]. The long edge and short edge of images are resized to 2048 and 512, respectively, without changing the aspect ratio (described as $512 \times 512$ in the main text for short). Then random crop, horizontal flip, and photometric distortion augmentations are adopted.

## B    Benchmark Results

### B.1    Instance Segmentation

**Accuracy comparison.** In Table 2 of the main text, we compare both accuracy and inference speed of K-Net with previous methods. For fair comparison, we re-implement Mask R-CNN [5] and Cascade Mask R-CNN [1] with the multi-scale $3\times$ training schedule [2, 12], and submit their predictions to the evaluation server[1] for obtaining their accuracies on the `test-dev` split. For SOLO [10], SOLOv2 [11], and CondInst [9], we test and report the accuracies of the models released in their official implementation [8], which are trained by multi-scale $3\times$ training schedule. This is because the papers [10, 11] of SOLO and SOLOv2 only report the results of multi-scale $6\times$ schedule, and the $AP_s$, $AP_m$, and $AP_l$ of CondInst [9] are calculated based on the areas of bounding boxes rather than instance masks due to implementation bug. The performance of TensorMask [3] is reported from Table 3 of the paper. The results in Table 2 show that K-Net obtains better $AP_m$ and $AP_l$ but lower $AP_s$ than Cascade Mask R-CNN. We hypothesize this is because Cascade Mask R-CNN rescales the regions of small, medium, and large objects to a similar scale of $28 \times 28$, and predicts masks on that scale. On the contrary, K-Net predicts all the masks on a high-resolution feature map.

**Inference Speed.** We use frames per second (FPS) to benchmark the inference speed of the models. Specifically, we benchmark SOLO [10], SOLOv2 [11], CondInst [9], Mask R-CNN [5], Cascade

---

[1]https://competitions.codalab.org/competitions/20796

Mask R-CNN [1] and K-Net with an NVIDIA V100 GPU. We calculate the pure inference speed of the model without counting in the data loading time, because the latency of data loading depends on the storage system of the testing machine and can vary in different environments. The reported FPS is an average FPS obtained in three runs, where each run measures the FPS of a model through 400 iterations [2, 12]. Note that the inference speed of these models may be updated due to better implementation and specific optimizations. So we present them in Table 2 only to verify that K-Net is fast and effective.

## B.2 Semantic Segmentation

In Table 3b of the main text, we compare K-Net on UperNet [13] using Swin Transformer [6] with the previous state-of-the-art obtained by Swin Transformer [6]. We first directly test the model in the last row of Table 3a of the main text (52.0 mIoU) with test-time augmentation and obtain 53.3 mIoU, which is on-par with the current state-of-the-art result (53.5 mIoU). Then we follow the setting in Swin Transformer [6] to train the model with larger scale, which resize the long edge and short edge of images to 2048 and 640, respectively, during training and testing. The model finally obtains 54.3 mIoU on the validation set, which achieves new state-of-the-art performance on ADE20K.

## C   Visual Analysis

**Masks Refined through Kernel Update.** We analyze how the mask predictions change before and after each round of kernel update as shown in Figure A1. The static kernels have difficulties in handling the boundaries between masks, and the mask prediction cannot cover the whole image. The mask boundaries are gradually refined and the empty holes in big masks are finally filled through kernel updates. Notably, the mask predictions after the second and the third rounds look very similar, which means the discriminative capabilities of kernels start to saturate after the second round kernel update. The visual analysis is consistent with the evaluation metrics of a similar model on the `val` split, where the static kernels before kernel update only achieve 33.0 PQ, and the dynamic kernels after the first update obtain 41.0 PQ. The dynamic kernels after the second and the third rounds obtain 46.0 PQ and 46.3 PQ, respectively.

**Failure Cases.** We also analyze the failure cases and find two typical failure modes of K-Net. First, for the contents that have very similar texture appearance, K-Net sometimes have difficulties to distinguish them from each other and results in inaccurate mask boundaries and misclassification of contents. Second, as shown in Figure A2b, in crowded scenarios, it is also challenging for K-Net to recognize and segment all the instances given limited number of instance kernels.

## D   Broader Impact

Simplicity and effectiveness are two significant properties pursued by computer vision algorithms. Our work pushes the boundary of segmentation algorithms through these two aspects by providing a unified perspective that tackles semantic, instance, and panoptic segmentation tasks consistently. The work could also ease and accelerate the model production and deployment in real-world applications, such as in autonomous driving, robotics, and mobile phones. The model with higher accuracy proposed in this work could also improve the safety of its related applications. However, due to limited resources, we do not evaluate the robustness of the proposed method on corrupted images and adversarial attacks. Therefore, the safety of the applications using this work may not be guaranteed. To mitigate that, we plan to analyze and improve the robustness of models in the future research.

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

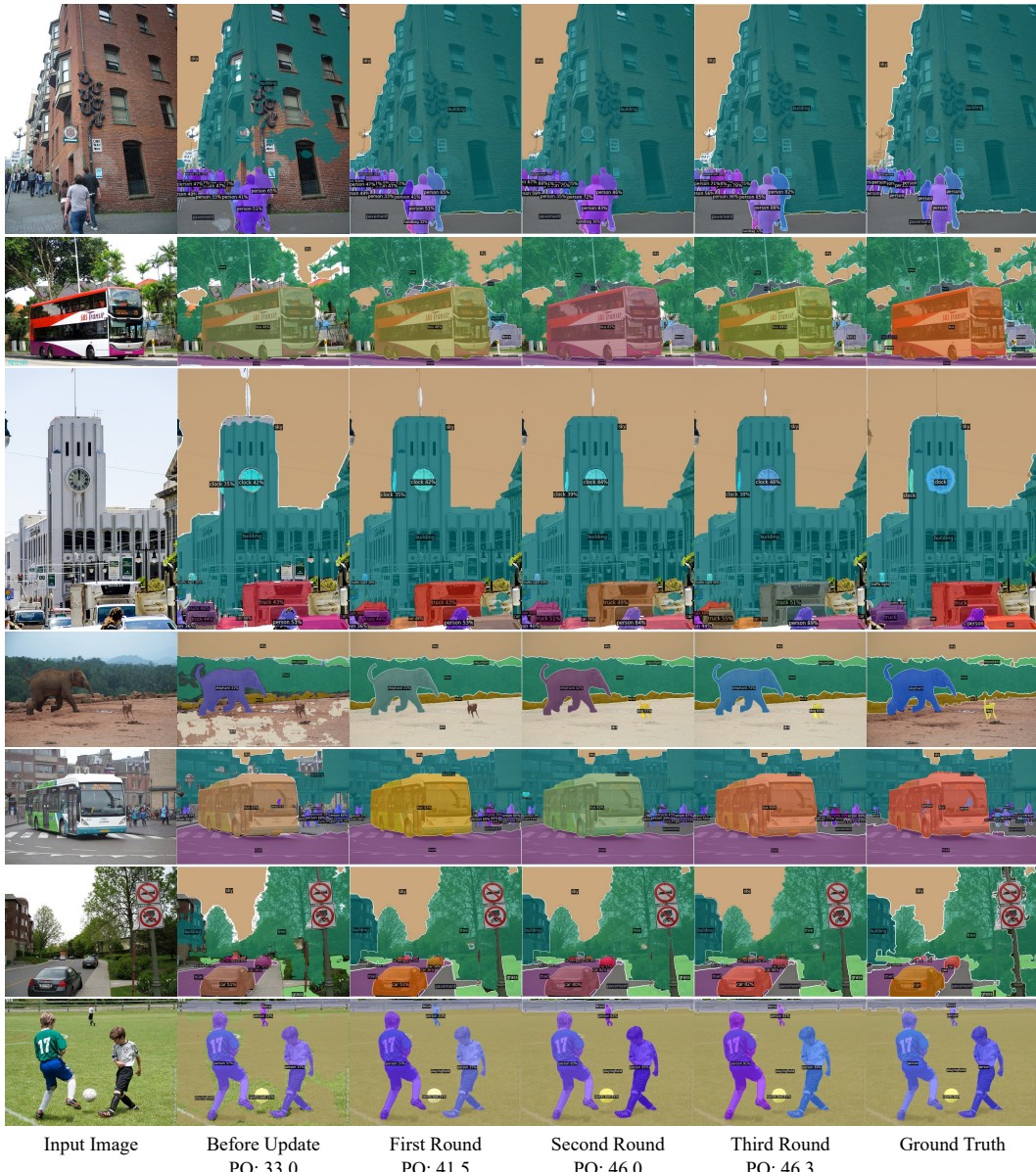

| Input Image | Before Update | First Round | Second Round | Third Round | Ground Truth |
|---|---|---|---|---|---|
| | PQ: 33.0 | PQ: 41.5 | PQ: 46.0 | PQ: 46.3 | |

Figure A1: Changes of masks before and after kernel updates of a similar model. PQ of each stage on the `val` split is also reported. Best viewed in color with zoom in.

[4] MMSegmentation Contributors. MMSegmentation: Openmmlab semantic segmentation toolbox and benchmark. `https://github.com/open-mmlab/mmsegmentation`, 2020.

[5] Kaiming He, Georgia Gkioxari, Piotr Dollár, and Ross B. Girshick. Mask R-CNN. In *ICCV*, 2017.

[6] Ze Liu, Yutong Lin, Yue Cao, Han Hu, Yixuan Wei, Zheng Zhang, Stephen Lin, and Baining Guo. Swin Transformer: Hierarchical vision transformer using shifted windows. *arXiv preprint arXiv:2103.14030*, 2021.

[7] Ilya Loshchilov and Frank Hutter. Decoupled weight decay regularization. In *ICLR*, 2019.

[8] Zhi Tian, Hao Chen, Xinlong Wang, Yuliang Liu, and Chunhua Shen. AdelaiDet: A toolbox for instance-level recognition tasks. `https://git.io/adelaidet`, 2019.

[9] Zhi Tian, Chunhua Shen, and Hao Chen. Conditional convolutions for instance segmentation. In *ECCV*, 2020.

[10] Xinlong Wang, Tao Kong, Chunhua Shen, Yuning Jiang, and Lei Li. SOLO: Segmenting objects by locations. In *ECCV*, 2020.

[11] Xinlong Wang, Rufeng Zhang, Tao Kong, Lei Li, and Chunhua Shen. SOLOv2: Dynamic and fast instance segmentation. *NeurIPS*, 2020.

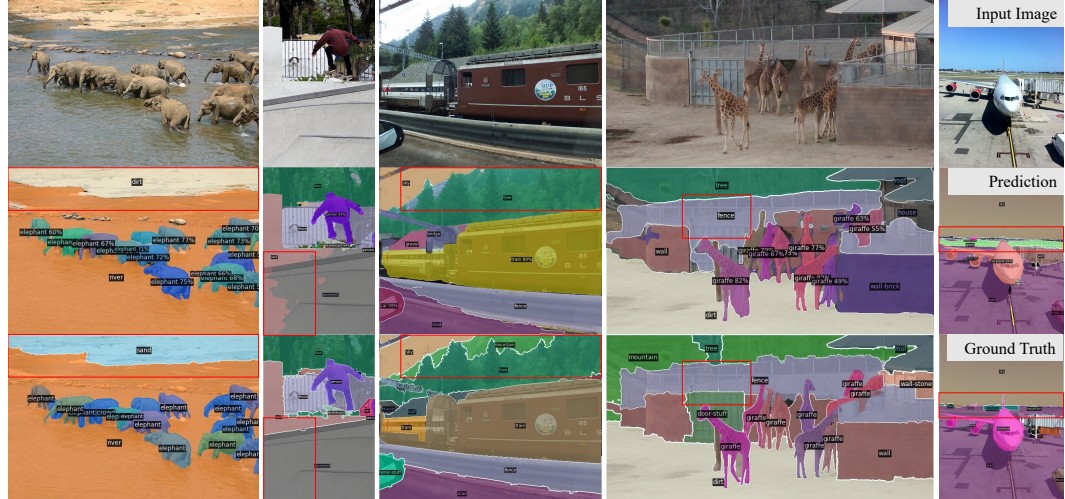

(a) Misclassification and inaccurate boundaries between contents that share similar texture appearance.

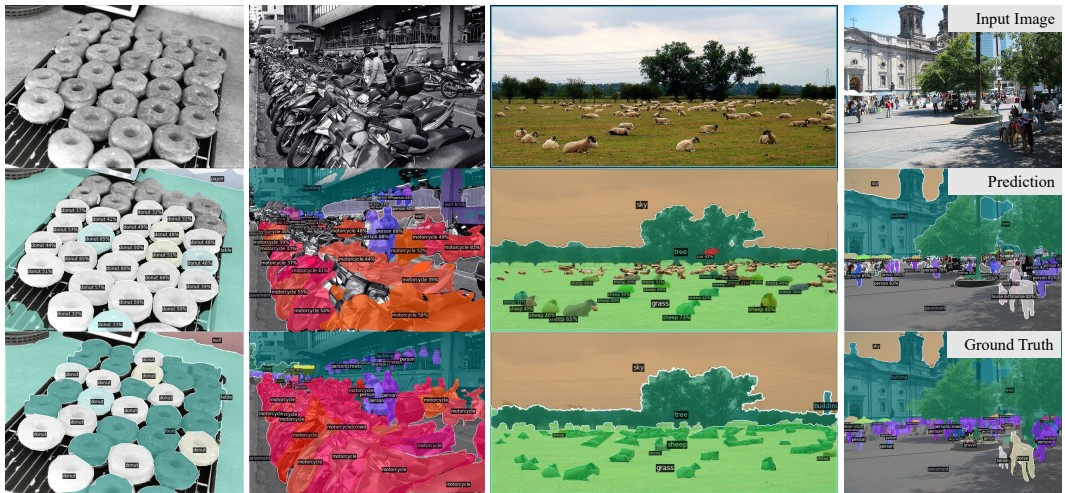

(b) Mask prediction on the images that contain crowded instances.

Figure A2: Failure modes of K-Net. Best viewed in color with zoom in.

[12] Yuxin Wu, Alexander Kirillov, Francisco Massa, Wan-Yen Lo, and Ross Girshick. Detectron2. `https://github.com/facebookresearch/detectron2`, 2019.

[13] Tete Xiao, Yingcheng Liu, Bolei Zhou, Yuning Jiang, and Jian Sun. Unified perceptual parsing for scene understanding. In *ECCV*, 2018.