# OpenReview forum: "K-Net: Towards Unified Image Segmentation"
_NeurIPS.cc/2021/Conference — NeurIPS 2021 Poster_

### Official Review · Reviewer_LzXD · 2021-07-05

**Rating:** 9
**Confidence:** 5

**Summary:**

This paper propose a unified framework named K-Net which segments both instances and semantic categories consistently by a group of learnable kernels, where each kernel is responsible for generating a mask for either a potential instance or a stuff class. The authors propose a refined kernel update strategy to group categories efficiently and effectively. Extensive experiments on three different segmentation tasks prove that such simple design can achieve STOA results while running faster than many previous hand-crafted segmentation methods.


**Limitations And Societal Impact:**

No major weakness on both method and experiment parts.

Here are several detailed suggestions and questions.

Suggestions:

1. Better cite and compare Max-Deeplab [1].
2. It seems that using kernel is a little confusing. Better to discuss the difference with object query in DETR [2] or Max-Deeplab [1].
3. Better give the detailed network architectures. (You can put it into supplementary just like Max-Deeplab).
4. What is g  in Line 182 (need the defination)?  I guess it is a fc layer.


[1] MaX-DeepLab: End-to-End Panoptic Segmentation with Mask Transformers CVPR-2021
[2] End-to-End Object Detection with Transformers ECCV-2020

**Main Review:**

Overall, from the motivation aspect, this is the most interesting paper in the topic of segmentation I have reviewed in this year. It brings a new way of modeling three segmentation tasks in one framework.  The new concept is to unify all the segmentation tasks which is also the goal of panoptic segmentation. This will be a fulture direction of segmentation task.

Although MaxDeeplab[1] and DETR[2] solve the panoptic segmentation in a NMS free manner, the former requires specific backbone and attention layers and the latter needs much longer time to converge also along with the heavy mask head.
K-Net can fairly compare with previous work using ResNet backbone (PanopticFCN, Panoptic FPN) and generalize well on large backbone (Swin-Transformer[3]).


Here are the more details:

1, Writing: The overall writing is good and easy to follow.

2, Method: The proposed framework is simple yet effective which has novel components such as kernel update for interative refine. The kernel fusion is updated by the  learnable gate which is reasonable.
Moreover such design is also computationally efficient which can be a plug-in module for other tasks such as human couting and pose estimation.

3, Experiment: The proposed K-Net surpasses all previous state-of-the-art single-model results of panoptic segmentation on MS COCO and semantic segmentation on ADE20K.  Most importantly, the comparison is fair and convincing.
The ablation studies are enough and good to prove the effectiveness of each part in method.

4, Potential: The framework is easy to extend into related tasks such as video segmentation tasks.

Thus I give my init rating as strong accept.
Hope the code can be available for the community soon.

[1] MaX-DeepLab: End-to-End Panoptic Segmentation with Mask Transformers CVPR-2021

[2] End-to-End Object Detection with Transformers ECCV-2020

[3] swin transformer: hierarchical vision transformer using shifted windows.

**Time Spent Reviewing:**

8.0

---

> ### Author Response · Authors · 2021-08-10
> **Reponse1 to Reviewer LzXD**
>
> **Q1:** Better cite and compare MaX-Deeplab [1].
>
> **A1:** We will cite and compare with it in the updated version. The concurrent work, MaX-DeepLab, focuses on performing panoptic segmentation with a transformer in an end-to-end manner. On the contrary, this paper aims to simplify instance segmentation by adopting the core design for semantic segmentation. K-Net directly applies a set of convolutional kernels to predict masks for different segmentation tasks, without relying on transformers nor any special losses. Such a framework tackles different segmentation tasks consistently through a unified perspective.
>
> Notably, K-Net outperforms Max-DeepLab on the COCO panoptic segmentation test-dev set by 0.8 PQ (52.1 vs. 51.3) with less training epochs.
>
> ------
> **Q2:** It seems that using kernel is a little confusing. Better to discuss the difference with object query in DETR [2] or Max-Deeplab [1].
>
> **A2:** We will update the paper and discuss that. The query in DETR or MaX-DeepLab is used to produce attention maps and to gather features for performing bounding box regression, classification, or mask prediction.
> However, in K-Net, convolutional kernels are used to perform convolution to directly obtain segmentation masks.
>
> ------
> **Q3:** Better give the detailed network architectures. (You can put it into supplementary just like Max-Deeplab).
>
> **A3:** Thanks and we will do that.
>
> ------
> **Q4:** What is $g$ in Line 182 (need the defination)? I guess it is a fc layer.
>
> **A4:** Yes, it is a nonlinear predictor containing FC-LN-ReLU-FC (FC and LN are fully connected and LayerNorm, respectively).

---

### Official Review · Reviewer_ukKk · 2021-07-10

**Rating:** 7
**Confidence:** 5

**Summary:**

This paper tries to unify semantic and instance segmentation with a group of learnable kernels that is conditioned on image features. Although the idea of using dynamic kernels (aka dynamic convolutions) has been already explored in instance segmentation [44, 47] and panoptic segmentation [30], this paper proposes to use sparse dynamic kernels instead of using kernels that are generated on dense grid in [30, 44, 37]. This paper further borrows the idea of bipartite matching loss in DETR [4] to make their model NMS free. The experiments show competitive results compared to state-of-the-art methods on multiple segmentation tasks.

**Limitations And Societal Impact:**

yes

**Main Review:**

First of all, I have concerns to the writing:

(1) In my optional, Table 4 (a) is a very important and also interesting experiments, however, there is not much description to this experiment. First, what is the baseline model without A.K.U. and K.I.? This is very important, because the improvements of these two components seem to be very significant. I want to make sure the author is not making comparison to some unreasonable baseline. Especially for without A.K.U., is kernel feature simply updated by $\tilde K = F^K + K_{i-1}$? If this is the case, why gating makes so much improvement? Another question is, without A.K.U., why K.I. makes performance worse? This does not make much sense, adding self-attention is possible to not improve the performance, but I don't expect it to decrease the performance.

(2) L295, positional encoding is mentioned here for the first time. It is unclear where positional encoding is added.

(3) L42, the authors claim they make two "novel" designs here. I would suggest the authors to not make a such strong claim. Especially for the second design: bipartite matching loss is proposed by DETR [4] and the authors should not claim it as their novel design.

Second, I have some questions/concerns on Adaptive Kernel Update (A. K. U.): According to the ablation study in Table 4 (a), the proposed A.K.U. is the most important component in K-Net. However, the motivation of A.K.U. is not very clear in my opinion. In the draft, the authors claim since $M_i$ is not precise, $F^K$ may contain noises and the gating mechanism can reduce the adverse effect of the noise. My questions are:
(1) Since there is no spatial resolution anymore, the only way to reduce the effect of noise is to reduce the amount of information from $F^K$ to $\tilde K$. But why the multiplication between $F^K$ and $K_{i-1}$ can provide such information?

(2) Why there is also a gate needed for $K_{i-1}$ as well?

(3) The proposed gating seems very simple according to the text description, so I have tried such gating myself with a simplified K-Net architecture (without semantic features on instance segmentation). However, I found the gating mechanism does not work as well as that is reported in Table 4 (a). In fact, it always decreases performance. I'm wondering if the authors provide the exact description on how they implement the A.K.U. and is there any special design that is not mentioned in the paper?

Some minor suggestions:

(1) L182: $g_i$ is undefined?

(2) L217: does it mean there is still conflict between instance and semantic segmentation so you need to paste instances first?

In general, this paper proposes an interesting architecture and performs enough ablation studies to validate the architecture. But the lack of motivation to the most important component (A.K.U.) and the difficulty to reproduce this important component lead to the weak accept score. If the authors can well address my concerns (especially on the A.K.U. part), I am happy to raise the score.




----
\#### Post rebuttal comment ####

The authors have addressed my concern on the reproduction issue. The new ablation Table is much more reasonable than the original one in the submission. On the other hand, the new ablation Table shows one of the proposed components is less effective and may not even be necessary.

As I said in my original review, this paper proposes an interesting architecture and it is of value to the community. Since the authors address my concerns, I will raise my score to something between 6 and 7 (more towards 7).

As pointed out by multiple reviewers, the presentation of this paper is not satisfactory: it misses many important details. Thus, I would suggest the authors to check their writing more carefully, and provide all the necessary context and details.

**Time Spent Reviewing:**

2 hours

---

> ### Author Response · Authors · 2021-08-10
> **Response to Reviewer ukKk**
>
>
> **Q1:** What is the baseline model without A.K.U. and K.I.? ... is kernel feature simply updated by $\tilde{K}=F^{K}+K_{i−1}$?
>
> **A1:** Yes, it is updated by $\tilde{K}=F^{K}+K_{i−1}$. The resulting features are directly used for kernel interaction if K.I is used.
>
> ------
> **Q2:** Without A.K.U., why K.I. makes performance worse? … I don't expect it to decrease the performance.
>
> **A2:** It decreases the performance because missing non-linear transformation after addition and before the multi-head attention makes the model hard to learn. We further examine the implementation in the experiments of Table 4(a). The baseline can be stronger if we add an FC-LN-ReLU layer after addition. We will update table 4(a) as below:
>
> | A.K.U | K.I.  | AP    | AP$_{50}$ | AP$_{70}$ |
> |:-----:|:-----:|:-----:|:---------:|:---------:|
> |       |       | 10	|18.2	    |9.6        |
> |Y      |       |22.6	|37.3	    |23.5       |
> |       | Y     |31.2	|52.0       |32.4       |
> |Y      |Y      |34.1	|55.3	    |35.7       |
>
> ------
> **Q3:** Why gating makes so much improvement?
>
> **A3:** The updated experiments in **Q2A2** show that gating brings less improvement than kernel interaction. It plays a role like the self-attention mechanism [1]. The output of self-attention is computed as a weighted summation of the values, where the weight assigned to each value is computed by a compatibility function of the query with the corresponding key. Similarly, adaptive kernel update essentially performs weighted summation of kernel features and group features, where their weight are computed by a compatibility function between the group features and kernel features in Eq 4.
>
> [1] Attention Is All You Need, NeurIPS2017.
>
> ------
> **Q4:** L295, positional encoding is mentioned here for the first time. It is unclear where positional encoding is added.
>
> **A4:** It is first mentioned in L202-203. Specifically, given the feature maps ${P2, P3, P4, P5}$ produced by FPN, positional encoding is computed based on the feature map size of $P5$, and it is added with $P5$. Then semantic FPN is used to produce the final feature map. We will clarify the details and release the code.
>
> ------
> **Q5:** I would suggest the authors to not make a such strong claim. Especially for the second design: bipartite matching loss is proposed by DETR [4] and the authors should not claim it as their novel design.
>
> **A5:** We agree and will tone down this claim.
>
> ------
> **Q6:** Since there is no spatial resolution anymore, the only way to reduce the effect of noise is to reduce the amount of information from $F^K$ to $\tilde{K}$. But why the multiplication between $F^K$ and $K_{i−1}$ can provide such information?
>
> **A6:** See **Q3A3**. The multiplication between $F^K$ and $K_{i−1}$ works like a compatibility function, similar to the dot-product used in self-attention mechanisms.
>
> ------
> **Q7:** Why there is also a gate needed for $K_{i−1}$ as well?
>
> **A7:** We add a gate of $K_{i-1}$ for symmetry referring to the design in LSTM [1] where both input gate and forget gate are important. It works slightly better.
>
> [1] An Empirical Exploration of Recurrent Network Architectures, ICML 2015.
>
> ------
> **Q8:** If the authors provide the exact description on how they implement the A.K.U. and is there any special design that is not mentioned in the paper?
>
> **A8:** The structure should be FC-LN-ReLU, LayerNorm (LN) is always used after the FC layer. We will update the paper
>
> ------
> **Q9:** L182: $g_i$ is undefined?
>
> **A9:** The function $g_i$ is an FC-LN-ReLU-FC layer, we will update it in the next version.
>
> ------
> **Q10:** L217: does it mean there is still conflict between instance and semantic segmentation so you need to paste instances first?
>
> **A10:** No, it does not mean that. For fair comparison and simplicity, we adopt a standard procedure similar to that in Panoptic FPN [1] and Panoptic FCN [2] to resolve the overlapped areas between masks.  We also tried another strategy based on a concurrent work MaskFormer [3], which pastes masks of things and stuff in a mixed order. It consistently improves K-Net by 0.3~0.4 PQ. It will be discussed in the updated version.
>
> [1] Panoptic feature pyramid networks, CVPR2019.
>
> [2] Fully Convolutional Networks for Panoptic Segmentation, CVPR2020.
>
> [3] MaskFormer: Per-Pixel Classification is Not All You Need for Semantic Segmentation.

---

> ### Author Response · Authors · 2021-08-27
> **Response to Post Comment of Reviewer ukKk**
>
> Thanks for the appreciation.
>
> The new ablation table shows that A.K.U is still necessary for high performance, but removing it will not cause the model's failure. We will further study that.
>
> We are revising the paper now. We will update all the necessary details and release code.

---

### Official Review · Reviewer_c5jK · 2021-07-16

**Rating:** 6
**Confidence:** 4

**Summary:**

The paper presents a convolutional neural network method for panoptic (multi-class, multi-instance) image segmentation. Semantic segmentation is addressed in the standard fashion with pre-trained model weights (convolutional kernels). Novelty is introduced in the instance segmentation component where the parameters of the convolutional kernels responsible for producing activation maps, one for each instance, are adapted at test-time to fit the given image. Overall this is a neat idea and the experimental results are compelling. However, I found the paper difficult to understand on first readings and have several suggestions for improvement of the presentation.

**Limitations And Societal Impact:**

I could not identify any societal impacts.

**Main Review:**

Overall I found the idea of adaptive model components for instance segmentation appealing but the presentation of the paper lacked clarity. It was not immediately clear (although obvious after reading Section 3) that the kernels referred to in the paper are filter kernels for convolutional neural networks. The term "kernel" has multiple meanings in the machine learning literature and this should be clarified much earlier in the paper.

My biggest concern relates to details of the initialization and updating of the adaptive instance kernels, which is the core contribution of the paper. First, it is not clear how the number, N, of kernels is chosen and what happens (at test time) if this number is over- or under-estimated. Second, it seems at least when initialized that the instance kernels should be more sensitive to spatial location than appearance, and then adapt to appearance. Can the authors comment? Also, is there a need to treat "thing" and "stuff" instances differently?

Third, it may be helpful to summarise the iterative updating of the kernels  in a formal algorithm. Eqn (2) gives the high-level overview although it is not clear why function $f$ is indexed by $i$. The spatial indices $(i,j)$ in Eqn (3) clash with the index $i$ for the iteration number and its not clear how the batches are indexed (presumably the operations are applied independently for each batch). Eqn (5) introduces $\tilde{K}$ without providing an explicit relationship to $K_i$ (from the following paragraph appear to be a transformer followed by multi-layer perceptron?). All of this can be presented more rigorously.

Last, it is not clear what effect the number of iterations applied to adapting the kernels has on the results. Table 3(c) suggests that accuracy saturates at four iterations for ADE20K and doesn't decrease thereafter but further data points should be given. Does the same finding hold for COCO? How many iterations (rounds) were used during training and does this affect the finding?

Some other minor things to address:
1. The image on page 1 is poorly placed and distracts the flow of the paper. I suggest moving to the top of the second page.
2. The idea of joint semantic and instance segmentation appears in the literature well before the term panoptic segmentation was proposed (line 20). For example, Tu et al., IJCV 2005 propose the idea of image parsing, Gould et al., NeurIPS 2009 propose the idea of image decomposition, and Yao et al., CVPR 2012 develop the idea of holistic scene understanding.

**Time Spent Reviewing:**

4

---

> ### Author Response · Authors · 2021-08-10
> **Response to Reviewer c5jK**
>
> **Q1:** It was not immediately clear … The term "kernel" … should be clarified much earlier in the paper.
>
> **A1:** Thanks for the suggestion. It refers to “convolutional kernels” and we will clarify it in the introduction.
>
> ------
>
> **Q2:** It is not clear how the number, $N$, of kernels is chosen and what happens (at test time) if this number is over- or under-estimated.
>
> **A2:** $N$ is selected considering the inference efficiency and the average number of objects in the image (7.7 for COCO dataset [1]).
> A larger $N$ may lead to small performance gains and then get saturated (when $N=300, 512, 768$, we all get 34.9 mAP), and a smaller one usually decreases the performance, as shown in Table 4(d).
>
> [1] Microsoft COCO: Common Objects in Context, ECCV2014.
>
> [2] End-to-End Object Detection with Transformers, ECCV2020.
>
> ------
> **Q3:** ...it seems at least when initialized that the instance kernels should be more sensitive to spatial location than appearance, and then adapt to appearance. Can the authors comment?
>
> **A3:** We wish to clarify Figure 3(a). The Figure 3(a) only shows how the kernels respond to spatial locations. We do not observe that the kernels first become sensitive to spatial location then adapt to appearance.
>
> ------
> **Q4:** Also, is there a need to treat "thing" and "stuff" instances differently?
>
> **A4:** It is unnecessary to treat ‘thing’ and ‘stuff’ masks differently to produce a reasonable performance, but we empirically find that generating initial masks of ‘thing’ and ‘stuff’ from $F^{ins}$ and $F^{sem}$ respectively yields better performance (about 1PQ).
>
> ------
> **Q5:** It may be helpful to summarise the iterative updating of the kernels in a formal algorithm
>
> **A5:** Thanks and we will revise it.
>
> ------
> **Q6:** Last, it is not clear what effect the number of iterations applied to adapting the kernels has on the results. Table 3(c) suggests that accuracy saturates at four iterations for ADE20K and doesn't decrease thereafter but further data points should be given. Does the same finding hold for COCO? How many iterations (rounds) were used during training and does this affect the finding?
>
> **A6:** We believe that you refer to Table 4 (c), of which the results are reported on COCO dataset. For efficiency, iteration number 3 is used for all experiments on COCO and ADE20K dataset if without further specification. The AP of 6 and 7 iterations on the COCO dataset is 34 mAP and 33.7 mAP, respectively. These results are lower than that of 4 & 5 iterations (34.5 mAP), indicating that the performance saturates starting from 4 iterations.
>
> The ablation study of iteration numbers on ADE20K is as below. The results also show that the performance also saturates after four iterations.
>
> | Stage Number  | mIoU  |
> |:-------------:|:-----:|
> |      0        | 36.7  |
> |      1        | 42.7  |
> |      2        | 43.0  |
> |      3        | 43.3  |
> |      4        | 43.8  |
> |      5        | 44.1  |
> |      6        | 43.1  |
> |      7        | 42.6  |
>
> ------
> **Q7:** Minor things to address.
>
> **A7:** Thanks for your kind suggestion, we will update Figure 1 and will discuss more previous works in the updated version.

---

> > ### Comment · Reviewer_c5jK · 2021-08-31
> > **Feedback**
> >
> > Thank you for your response. I am still leaning towards acceptance of this paper.

---

### Official Review · Reviewer_PzxM · 2021-07-18

**Rating:** 6
**Confidence:** 4

**Summary:**

This paper presents a unified approach for Semantic, instance, and panoptic segmentations tasks.  Proposed method segments both instances and semantic categories consistently by a group of learnable kernels, where each kernel is responsible for generating a mask for either a potential instance or a stuff class.
The model is training using bipartite matching, and the training and inference are naturally NMS-free and box-free.


**Ethical Concerns:**

I don't have any concern regarding ethical concern.

**Limitations And Societal Impact:**

The discussion about broader impact look good to me.

**Main Review:**

The panoptic segmentation approaches usually use a separate method for prediction things and stuff. It’s nice that this paper proposes a unified approach for them.

The proposed method is evaluated on three tasks of coco panoptic segmentation, coco instance segmentation and ADE20k semantic segmentation.
They show that by adding k-net on top of different backbone and model architectures they can consistently improve the performance of the model.
On the coco panoptic segmentation task they could achieve a new state-of-the-art result.
However, the best result on coco instance segmentation is much lower than state-of-the-art. Also the improvement on instance segmentation task seems to be less significant. I’m wondering why is that? Does k-net give more improvement on stuff compared to things?
It would be nice to provide per category improvements by adding k-net to one of the models for the panoptic segmentation task..


**Time Spent Reviewing:**

3

---

> ### Author Response · Authors · 2021-08-10
> **Response to Reviewer PzxM**
>
> **Q1:** The best result on coco instance segmentation is much lower than state-of-the-art. Also the improvement on instance segmentation task seems to be less significant. I’m wondering why is that?
>
> **A1:** First, we believe that K-Net is not worse than state of the art. K-Net achieves comparable performance with Cascade Mask R-CNN with nearly 2$\times$ faster inference speed (19.8 vs. 10.3 FPS, first row in Table 2). On the contrary, many recent approaches like SOLOv2 [1] and CondInst [2] cannot even surpass Cascade Mask R-CNN and are slower than K-Net.
>
> Second, some state-of-the-art methods adopt more complicated structures to enhance the instance segmentation performance. For example, SCNet [3] extends Cascade Mask R-CNN by global context encoding with stuff-thing annotations in the semantic segmentation branch. K-Net can also be improved by adding those general techniques, which is beyond the scope of this paper and we leave it to future research.
>
> We wish to emphasize that the primary goal of this paper is to use a framework as simple as that in semantic segmentation to tackle instance and panoptic segmentation tasks. Our work simplifies the instance segmentation framework by the novel perspective of kernels, which also tackles different segmentation tasks consistently from a unified perspective.
>
> [1] SOLOv2: Dynamic and Fast Instance Segmentation, NeurIPS2020.
>
> [2] Conditional Convolutions for Instance Segmentation, ECCV2020.
>
> [3] SCNet: Training Inference Sample Consistency for Instance Segmentation, AAAI2021.
>
> ------
>
> **Q2:** Does k-net give more improvement on stuff compared to things? It would be nice to provide per category improvements by adding k-net to one of the models for the panoptic segmentation task.
>
> **A2:** From Table 1, K-Net brings more significant improvement on PQ of stuff than PQ of things.
> We compare PQ of things and stuff classes between K-Net and Panoptic FCN with the ResNet-50 backbone. The results show that K-Net performs consistently better than Panoptic FCN on stuff classes, but may performs worse than Panoptic FCN on some thing classes. We select the PQ of thing and stuff classes in the Table below (better results are bolded). We will add a new section in the appendix to discuss that.
>
> *PQ of selected stuff classes*
>
> | Class names   | K-Net | Panoptic FCN  |
> |:-------------:|:-----:|:-------------:|
> | river         | **54.406**|47.106         |
> | road          | **60.789**|58.101         |
> | roof          | **17.059**|13.432         |
> | sand          | **56.358**|54.371         |
> | sea           | **76.718**|73.818         |
> | shelf         | **20.980**|19.074         |
> | sky           | **84.679**|83.098      |
>
> *PQ of selected thing classes*
>
> | Class names   | K-Net | Panoptic FCN  |
> |:-------------:|:-----:|:-------------:|
> | person        | 61.376| **65.022**    |
> | bicycle       | 39.543| **39.772**    |
> | motorcycle    | **55.117**| 52.898        |
> | train         | **74.592**| 72.397        |
> | truck         | **48.118**| 46.279        |
> | cat           | **79.712**| 79.557        |
> | dog           | **73.064**| 68.181        |

---

### Author Response · Authors · 2021-08-10
**Response to All Reviewers**

We thank all the reviewers for their valuable comments. We will update the draft according to the suggestions on the writing and figures. For specific questions of each reviewer, we answer them in the reply to each reviewer.

---

### Decision · Program_Chairs · 2021-09-27

**Decision:**

Accept (Poster)

**Comment:**

The submission discusses how to unify methods for semantic, instance and panoptic segmentation. For this the authors propose to use a group of learnable kernels and a corresponding update strategy. All reviewers appreciated the contributions of this paper and recommend acceptance/leaning to accept after having read each others reviews and the rebuttal, which addressed some concerns about missing/incomplete ablations. However, all reviewers also pointed out that presentation and writing are not satisfactory and lack clarity. AC read the paper and concurs. E.g., "With the constructed M_0, K_0, and F, the kernel update head can start produce group-aware kernels K_s, s = {1, 2, ..., S} iteratively by S times and obtain the refined mask prediction M_S." (randomly picked sentence). Sentences like this are hard to parse and may prevent others from replicating the results. This being said, the authors promised to release code and models.